# Multidisciplinary Tumor Board in the Management of Patients with Colorectal Liver Metastases: A Single-Center Review of 847 Patients

**DOI:** 10.3390/cancers14163952

**Published:** 2022-08-16

**Authors:** Flavio Milana, Simone Famularo, Antonio Luberto, Lorenza Rimassa, Marta Scorsetti, Tiziana Comito, Tiziana Pressiani, Ciro Franzese, Dario Poretti, Luca Di Tommaso, Nicola Personeni, Marcello Rodari, Vittorio Pedicini, Matteo Donadon, Guido Torzilli

**Affiliations:** 1Department of Biomedical Sciences, Humanitas University, 20072 Pieve Emanuele, MI, Italy; 2Department of Hepatobiliary and General Surgery, IRCCS Humanitas Research Hospital, 20089 Rozzano, MI, Italy; 3Humanitas Cancer Center, IRCCS Humanitas Research Hospital, 20089 Rozzano, MI, Italy; 4Radiotherapy and Radiosurgery Department, IRCCS Humanitas Research Hospital, 20089 Rozzano, MI, Italy; 5Department of Radiology, IRCCS Humanitas Research Hospital, 20089 Rozzano, MI, Italy; 6Department of Pathology, IRCCS Humanitas Research Hospital, 20089 Rozzano, MI, Italy; 7Department of Nuclear Medicine, IRCCS Humanitas Research Hospital, 20089 Rozzano, MI, Italy

**Keywords:** multidisciplinary team meeting, colorectal cancer, colorectal liver metastases

## Abstract

**Simple Summary:**

A multispecialty discussion may represent a benefit for cancer patients in receiving the most updated pathway of cure. The aim of this study was to describe the activity of our multidisciplinary team (MDT) meeting on patients with colorectal liver metastases (CRLMs). Information about 847 consecutive CRLMs patients over an 11-year period was retrospectively collected. Their characteristics were analyzed, and the populations were compared based on received treatment.

**Abstract:**

There is still debate over how reviewing oncological histories and addressing appropriate therapies in multidisciplinary team (MDT) discussions may affect patients’ overall survival (OS). The aim of this study was to describe MDT outcomes for a single cancer center’s patients affected by colorectal liver metastases (CRLMs). From 2010 to 2020, a total of 847 patients with CRLMs were discussed at our weekly MDT meeting. Patients’ characteristics and MDT decisions were analyzed in two groups: patients receiving systemic therapy (ST) versus patients receiving locoregional treatment (LRT). Propensity-score matching (PSM) was run to reduce the risk of selection bias. The median time from MDT indication to treatment was 27 (IQR 13–51) days. The median OS was 30 (95%CI = 27–34) months. After PSM, OS for patients undergoing LRT was 51 (95%CI = 36–64) months compared with 15 (95%CI = 13–20) months for ST patients (*p* < 0.0001). In this large retrospective study, the MDT discussions were useful in providing the patients with all available locoregional options.

## 1. Introduction

Colorectal cancer (CRC) is the third most common tumor and a leading cancer-related cause of mortality worldwide [1]. Almost 50% of patients diagnosed with CRC develop liver metastases [2,3]. Indeed, the liver represents the most common spreading site of metastatic colorectal disease, and one in four patients with this disease experiences synchronous colorectal liver metastases (CRLMs) [4].

The current guidelines for CRLMs treatment include the use of systemic and locoregional therapy, which both should be administered in a tailored fashion according to patient- and tumor-related features, aiming to achieve a 5-year survival rate of 60% [5,6,7,8]. To identify the optimal treatment strategy for CRLMs patients, every case should be discussed by a multidisciplinary team (MDT) of experts, taking into consideration the age, the general clinical conditions, the residual organ function, and the concomitant non-tumor diseases, defining the best therapeutic approach. Such an MDT approach is very important for those patients amenable to liver resection. Indeed, surgery represents a potential curative option associated with overall survival (OS) rates at 5 and 10 years of about 50% and 35%, respectively [9]. Although hepatic resection techniques for CRLMs have improved in recent decades [10], only a minority of patients who develop CRLMs could be candidates for upfront surgery, meaning the performance of hepatic surgery without preoperative chemotherapy; moreover, disease recurrence occurs in about 70% of these patients [11]. On the other hand, up to 25% of patients with a diagnosis of inoperable CRLMs become resectable after protocols of conversion systemic therapy [5,12]. 

Recently, increasing attention has been focused on multimodal treatment strategies, meaning that the single or repeated applications of locoregional and systemic therapies are considered potentially more effective in providing long-term survival than chemotherapy alone [13,14]. Indeed, good results have been reported using ablative techniques or transarterial therapies, which, as an alternative to or in combination with systemic therapy for CRLMs patients [15,16,17,18], can also be applied to patients with CRLMs recurrence after a given previous locoregional treatment [19,20].

In such a complex scenario, the role of an MDT should be the coordination and delivery of the appropriate cancer care, considering all the different therapeutic options that can be combined to offer the best chance of cure [21]. However, while the literature suggests that the MDT could be helpful in revising tumor stage and diagnosis, modifying management plans, and improving the adherence to clinical guidelines, its role in managing such new multimodality curative approaches is less clear [22,23,24,25].

The present study summarizes the experience of a single high-volume cancer center with the purpose of describing how the MDT is composed, how it works, and how CRLMs patients may be managed with a multimodality approach.

## 2. Materials and Methods

### 2.1. Study Protocol

This is an observational retrospective study on data collected consecutively and prospectively in a single high-volume Italian cancer center (IRCCS Humanitas Research Hospital, Rozzano, Milan, Italy). The results are reported according to the principles of Strengthening the Reporting of Observational Studies in Epidemiology (STROBE) [26]. All procedures were performed in accordance with the Helsinki Declaration and our institution’s ethical committee of Humanitas Research Hospital. All consecutive adult patients (age ≥18 years), discussed at the local MDT meeting with a diagnosis of CRLMs from January 2010 to December 2020, were evaluated. The inclusion criteria were the following: (1) radiological or histological diagnosis of CRLM and (2) available follow-up data. The exclusion criteria were: (1) lesions that were investigated by histology for any reason and not confirmed as CRLM and (2) missing data on the MDT therapeutic indications or follow-up. 

### 2.2. Definitions and Follow-Up Protocol

The Eastern Cooperative Oncology Group–Performance Status (ECOG-PS) was measured at the first outpatient visit [27,28]. Biochemical values were obtained within two weeks from the assigned treatment. The number and size of nodules were assessed by expert and dedicated hepatobiliary radiologists through multiphase contrast computed tomography (CT) or magnetic resonance imaging (MRI). The extension of liver resection was defined as minor or major, based on the Brisbane nomenclature [29], while post-hepatectomy complications were recorded by using the Clavien–Dindo classification [30]. For the purpose of the present study, only patients deemed to be resectable were included, meaning that systemic chemotherapy was, in fact, completed with perioperative intent. For these patients, the tumor response to chemotherapy was evaluated according to Response Evaluation Criteria in Solid Tumors (RECIST) v.1.1 [31]. All patients received follow-up using local protocols, which included measurement of serum tumor markers (carcinoembryonic antigen (CEA) and carbohydrate antigen (CA) 19-9, abdominal ultrasound, CT or MRI, and outpatient visits. OS was defined as the time from the date of the assigned treatment to any cause of death. Patient surveillance was closed at the end of December 2021.

### 2.3. Multidisciplinary Tumor Board

The MDT board included liver surgeons, hepatologists, medical oncologists, radiologists, interventional radiologists, radiation oncologists, and pathologists. When requested, thoracic surgeons and nuclear medicine physicians were also included. The board’s decisions considered several factors, such as the patient’s age, the underlying liver function, the tumor burden (size and number of lesions, unilobar vs. bilobar disease, mutational status), the anticipated residual liver volume after resection, the comorbidities, the previous surgical and medical history, and the patient’s preferences. In the era of precision medicine approaches, each case was evaluated based on the most recent scientific guidelines that, together with the local experience in managing patients with CRLMs, allowed us to offer the most appropriate patient-tailored treatment. The therapeutic options included: systemic therapy, surgical resection, thermoablation, transarterial therapy, and radiotherapy. Notably, the locoregional treatments were chosen when deemed to be definitive and effective treatments.

### 2.4. Data Presentation and Study Endpoints

The primary endpoint was to describe the whole population of patients with CRLMs observed and managed during the study period. The secondary endpoint was to derive a prognostic model of OS based on tumor and patient characteristics, such as age, sex, primary tumor stage classification (according to the AJCC staging system [32]), size and number of CRLMs, and type of treatment received together with the strategy adopted by the MDT board. The tertiary endpoint was to analyze patients treated with systemic therapy (ST) alone versus those treated with locoregional therapy (LRT), i.e., surgery and/or ablation. For this last analysis, propensity-score matching (PSM) was run to reduce the risk of selection bias [33]. Notably, OS was chosen as an outcome measure because it was considered to be the most reliable and unbiased endpoint in the research on CRLMs. In such a large patient population, it has been assumed that causes of death other than tumor recurrence would not have been affected by the variables chosen for the study. 

### 2.5. Statistical Analysis

The data were checked for normality by using the Kolmogorov–Smirnov test. The sample was described with the median and interquartile range (IQR) for numeric variables and the number and proportion for categorical variables. The Mann–Whitney U test and Fisher’s exact test were used to compare baseline patient characteristics between the two treatment groups. All the significant preoperative variables between treatment groups were employed to perform a 1:1 nearest neighbor PSM with a caliper of 0.1 standard deviation (SD) to balance the differences and to reduce the risk of selection bias among treatments. The following variables were considered for the PSM: the primitive pathological tumor staging (pT) and the primitive pathological nodal staging (pN), according to AJCC staging system [32]; synchronous versus metachronous liver metastases presentation; localization of the liver lesions (unilobar versus bilobar); presence of extrahepatic concomitant spread; number and size of liver lesions; and CEA. Loveplots were created to visualize the efficacy of matching for the considered variables. Survival was estimated by the Kaplan–Meier method, and comparisons were made by using the Log-rank test. The risk of overall mortality was estimated by Cox regression in the whole cohort and in the subgroup analysis, and the results were expressed as hazard ratio (HR) and 95% confidence interval (CI). All statistical tests were two-tailed, and a 5% significance level was adopted. All the analyses were computed by using open-source R software (v4.0.5). 

## 3. Results

### 3.1. Patient Baseline Characteristics

From January 2010 to December 2020, 2294 patients were discussed at the MDT meetings. Of these, 847 (36.9%) were affected by CRLMs. The number of cases per year and MDT treatment indications are presented in Figure 1a. The baseline characteristics of this cohort of patients are described in Table 1. Five hundred forty-nine patients (64.8%) were male. The median age at the time of MDT discussion was 63.00 years (IQR 55.00–71.00). The ECOG-PS was 0 for 566 (66.8%) patients. The main site of the primary CRC was the left colon (n = 340 (40.1%)), followed by rectal cancer (n = 227 (26.8%)). According to the histological examination, the primary tumor was staged T4 in 141 (16.6%) cases. KRAS was mutated in 325 (38.4%) cases, despite the mutational status being unknown in about 20% of the population. In 733 (86.5%) patients, the diagnosis was reached by using MRI with liver-specific contrast, and positron emission tomography (PET) was employed in 703 (83.0%) patients. Metachronous CRLMs were diagnosed in 318 (37.5%) patients. Four hundred ninety-six (58.6%) patients presented with bilobar disease. Extrahepatic concomitant disease was evident in 245 (28.9%) cases.

### 3.2. Patient Allocation Based on the MDT Decision

Among all patients discussed by the MDT with diagnoses of CRLMs, 521 (61.5%) patients underwent surgical resection, 250 (29.5%) patients were treated with chemotherapy alone, 21 (2.5%) patients received ablation, 31 (3.7%) patients underwent stereotactic-body radiation therapy (SBRT), 7 (0.8%) patients received transarterial chemoembolization, and 17 (2%) patients received best supportive care only (Figure 1b). The median time from MDT indication to treatment was 27 days (IQR 13–51); in particular, the time to treatment (TTT) was 17 days (IQR 7–27) for chemotherapy alone and 29 days (IQR 15–55) for liver resection. 

Among the 521 resected patients, perioperative chemotherapy was suggested for 452 patients, with a total of 404 patients effectively resected, while the upfront strategy was used in 117 patients. According to RECIST criteria, among the patients treated with preoperative chemotherapy and then surgery, only 1 (0.2%) had a complete radiological response, 287 (71%) had a partial response, 69 (17%) had stable disease, and 47 (11.8%) had progressive disease. Of note, the progression was considered to be minimal (dimensional and not numerical) and therefore, they were surgically resected. 

### 3.3. Survival Analysis and Risk Factors for Overall Survival

After a median follow-up time of 36 (95%CI = 18–59) months, the median OS for the whole cohort was 30 (95%CI = 27–34) months (Figure 2a). Patients who received surgery had a median OS of 43 (95%CI = 37–51) months, while those who were treated with chemotherapy alone had a median OS of 14 (95%CI = 13–17) months. In patients treated with SBRT, the median OS was 23 (95%CI = 21–NA) months, while it was 58 (95%CI = 30–NA) months in patients undergoing ablation. Figure 2b indicates the OS for each treatment. In the Cox multivariate regression analysis, a KRAS mutated tumor (HR 0.72, 95%CI = 0.54–0.96, *p* = 0.024), a curative treatment compared to chemotherapy alone (HR 0.21, 95%CI = 0.14–0.29, *p* < 0.001), a N2 CRC (HR 1.71, 95%CI = 1.20–2.43, *p* = 0.003), larger size of the liver metastases (HR 1.11, 95%CI = 1.06–1.17, *p* < 0.001), and elevated CEA (HR 1.00, 95%CI = 1.0–1.0, *p* = 0.050) were independent predictors of survival (Table 2). 

### 3.4. Systemic versus Locoregional Therapy: Propensity-Score Matching

In our cohort of 847 patients, 250 (29.5%) patients were treated with chemotherapy alone, while 542 (63.9%) patients underwent liver resection or ablation. As expected, the patients treated with chemotherapy rather than surgery or ablation had different baseline characteristics. In particular, the group of patients treated with surgery or ablation had a higher frequency of advanced colorectal tumor stage (T3–T4) (*p* < 0.001), a higher frequency of positive lymph nodes (N+) (*p* < 0.001), and a higher rate of metachronous disease (*p* = 0.029). The group of patients treated with chemotherapy alone had more frequent bilobar presentation of the disease (*p* < 0.001), a higher frequency of concomitant extrahepatic spread (*p* = 0.001), a higher number of hepatic nodules (5.00 [2.00–10.00] vs. 4.00 [2.00–7.00], *p* < 0.001), larger nodules (3.50 cm [2.20–5.60] vs. 3.00 cm [1.80–4.50], *p* < 0.001), and higher CEA levels (19 ng/mL [5.00–69.50] vs. 5.60 ng/mL [3.00–18.00], (*p* < 0.001). With the aim to balance these significant differences, 1:1 propensity-score matching analysis was applied and 418 patients were extracted, with 209 in each group. The baseline characteristics of these two groups before and after propensity-score matching are summarized in Table 3. Other data regarding patients submitted to chemotherapy alone are reported in Appendix A (Table A1). After the matching process, the median OS was 15 (95%CI = 13–20) months and 51 (95%CI = 36–64) months for those patients treated with chemotherapy and surgery or ablation, respectively (*p* < 0.001). Survival curves before and after weighting are depicted in Figure 3a,b.

## 4. Discussion

Although this is a retrospective investigation, it represents one of the largest single-center studies analyzing the multidisciplinary management of patients affected by CRLMs. Recently, increasing attention has been given to the beneficial role of a multispecialty planned discussion together with the importance of having a physical place where conflicting views can be discussed and complementary experiences can be integrated [22]. Nevertheless, limited data have been published so far on the impact of the different therapies addressed during MDT meetings [34]. The literature indicates that MDT discussions provide a forum to review imaging results, to change the diagnosis and staging of the disease, and to share information and responsibilities [35,36,37]. 

Our analysis represents an 11-year population study, and its strength lies in the large sample size (847 patients with stage IV CRC) and in the reliance on a single surgical, oncological, and radiological team to whom all patients with CRLMs were referred. The analysis shows an increasing number of patients discussed each year (up to 150 patients/year pre-COVID-19 pandemic), attesting an increasingly central role of the MDT meetings in the history of disease. In such a history, liver resection represents the standard curative treatment for patients with CRLMs whenever surgery is feasible, even in the most advanced clinical presentations [38,39]. One finding that has emerged from our data: almost two-thirds of the patients received multimodal treatment, including hepatectomy, with an associated median OS of 30 months. Another interesting point to make regards the TTT. Our data revealed that the median waiting time for surgery or for systemic therapy was 4 and 2 weeks, respectively, from the MDT board decision, indicating overall good management for CRLMs patients awaiting treatments. 

Such a large cohort of patients gave us the opportunity to analyze, over an 11-year period, the results associated with MDT choices. The median follow-up time of 36 months (increasing to 43 months for surgical patients) made comparisons possible, leading to outcomes that correspond with most of the data presented in the literature. Most of the candidates for surgical resection had perioperative systemic chemotherapy, as recommended by the current guidelines, and they were reconsidered for surgery only in case of radiological and biological response [5,6]. The same indication was also applied to patients thought to be easily resectable if they had bad oncological prognostic criteria. Indeed, 72% of patients had T3–T4 primary CRC, and 63% had synchronous metastatic disease, with 4 as the median number of CRLMs. Consistently, at the risk factors analysis for OS, a KRAS mutated tumor, an N2 CRC, a larger size of the liver metastases, and elevated preoperative CEA were found to be independently associated with worse survival. Of note, a curative treatment compared to chemotherapy alone was also found to be a predictor of survival, while the number of CRLMs, which are often considered prognostically relevant, was not significant. 

To further investigate the role of curative treatments versus chemotherapy, we decided to compare these two groups by using a matching process. As shown, patients treated with locoregional treatments had three-fold the survival rate of those patients treated with chemotherapy alone. Certainly, this survival rate is the result of the combination of appropriate patient selection and of the synergic effect of curative strategies pursued in a multimodality and integrated approach. However, it is true that surgical resection of CRLMs should be applied to ensure long-term survival, and in this sense, the multimodality approach should be that approach capable of downsizing the tumor burden with the intent of applying locoregional treatments whenever feasible.

Despite our results, evaluating MDT is challenging [13,22,36]. Based on the literature, MDT meetings have continued to improve, although a precise definition of what the meeting entails and who should be included is still needed [5,24]. Recently, guidelines for an effective MDT have been proposed, considering the necessity of standardizing the processes [40,41]. Of note, Keating et al. [42] emphasized the importance of the participants’ expertise as well as of the structural and functional components. When evaluating multidisciplinary care, another outcome to be measured is related to the costs of this activity both in terms of human resources and time [43,44]. While such costs are difficult to detail, it is our opinion that they are counterbalanced by the more defined clinical pathway provided by the MDT board for each patient. This probably could reduce other costs for healthcare professionals and patients [13,21,22,36]. Moreover, some authors do not support the concept of MDT, reporting how these multidisciplinary meetings are entities without well-defined moral and medicolegal responsibility, imputable of taking the worst decisions without personally dealing with the patients [45,46,47]. What is clear is that MDT discussions are valuable for complex cases where management is not straightforward and different therapeutic options carried out by different specialists may be applied.

The present study has several limitations that need to be considered. Due to the retrospective nature of this study, the results may be affected by biases in retrieving complete information about patients. While the potential selection bias was mitigated by the employment of propensity score matching, some intrinsic biological and clinical differences among treatment groups might have remained. Additionally, due to the low number of cases in some treatment groups, we could not exclude the risk of type-II error. Finally, the lack of a control group of patients managed without MDT discussion foreclosed any analysis on MDT effects on survival.

## 5. Conclusions

In conclusion, the management of CRLM patients requires MDT discussion for risk stratification and therapeutic approach, particularly for complex cases. While it is difficult to demonstrate the survival benefit provided by MDT meetings, it seems clear that the combination of strategies, which is best determined via a multispecialty and multimodality approach, represents a fundamental tool to provide the best achievable survival.

## Figures and Tables

**Figure 1 cancers-14-03952-f001:**
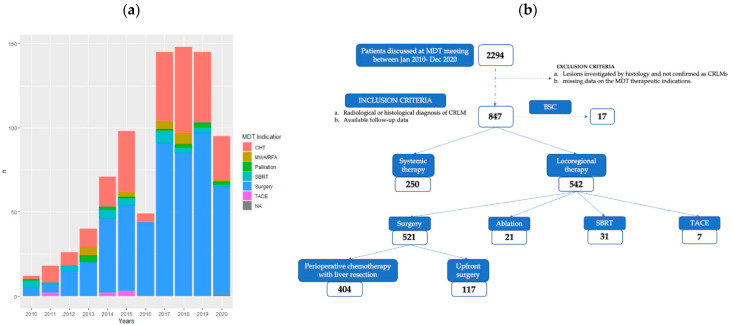
(**a**) Cases per year discussed with MDT treatment indication from January 2010 to December 2020. (**b**) Flowchart showing patient-selection process and MDT treatment indications (CHT: chemotherapy, MWA: microwave ablation, RFA: radiofrequency ablation, SBRT: stereotactic-body radiation therapy, TACE: transarterial chemo-embolization, BSC: best supportive care, NA: not available).

**Figure 2 cancers-14-03952-f002:**
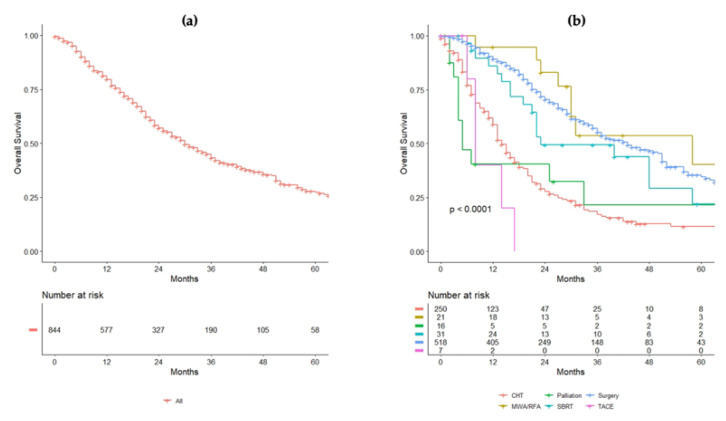
(**a**) Overall survival (OS) of the entire cohort. (**b**) OS of the population analyzed for each treatment received.

**Figure 3 cancers-14-03952-f003:**
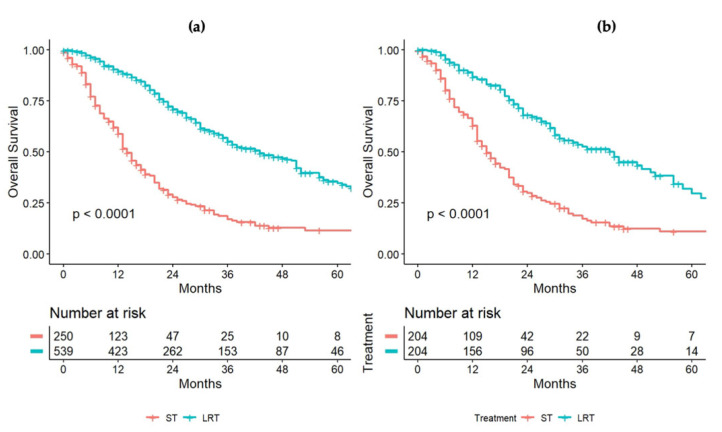
OS comparison between patients undergoing systemic therapy (ST) or locoregional treatment (LRT: surgery/thermablation), before (**a**) and after (**b**) the propensity-score-matching process.

**Table 1 cancers-14-03952-t001:** Baseline characteristics of the cohort of patients analyzed.

Variable	Overall Series
Number	847
Male sex (%)	549 (64.8)
Age (median; IQR)	63.00; 55.00, 71.00
ECOG PS (%)	
0	566 (66.8)
1	227 (26.8)
2	49 (5.8)
3	5 (0.6)
Site of primary tumor (%)	
ileum	2 (0.2)
cecum	59 (7.0)
ascending colon	168 (19.8)
transversal colon	51 (6.0)
descending colon	340 (40.1)
rectum	227 (26.8)
T stage (%)	
T 1-2	110 (12.9)
T 3-4	610 (72.1)
NA	127 (15.0)
N status (%)	
0	220 (26.0)
1	286 (33.8)
2	209 (24.7)
NA	132 (15.6)
KRAS mutational status (%)	
Mutated	325 (38.4)
Wild type	352 (41.6)
NA	170 (20.1)
NRAS mutational status (%)	
Mutated	42 (5.0)
Wild type	462 (54.5)
NA	343 (40.5)
BRAF mutational status (%)	
Mutated	22 (2.6)
Wild type	441 (52.1)
NA	384 (45.3)
Disease staging	
Computed tomography (%)	842 (99.4)
Magnetic resonance imaging (%)	733 (86.5)
Positron emission tomography (%)	703 (83.0)
Metachronous disease (%)	318 (37.5)
Liver bilobar disease	496 (58.6)
Size of the largest metastasis (cm) (median; IQR)	3.00; 1.90, 5.00
Number of metastases (median; IQR)	4.00; 2.00, 8.00
Extrahepatic disease (%)	245 (28.9)
CEA (ng/mL) (median; IQR)	7.00; 3.00, 30.00
MDT response (%)	
Surgery	521 (61.5)
Systemic therapy	250 (29.5)
SBRT	31 (3.7)
MWA/RFA	21 (2.5)
Palliation	17 (2)
TACE	7 (0.8)
Perioperative chemotherapy	452 (53.3)
N of perioperative cycles (median; IQR)	5.00; 0.00, 10.00
RECIST (%) *	
CR	1 (0.2)
PR	287 (63.5)
SD	69 (15.3)
PD	88 (19.5)
NA	7 (1.5)
Time to treatment (months) (median; IQR)	27.00; 13.00, 51.00

* Percentage calculated for patients receiving perioperative chemotherapy. (ECOG PS: Eastern Cooperative Oncology Group Performance Status, CT scan: computed tomography scan, MRI: magnetic resonance imaging, PET: positron emission tomography, CEA: carcinoembryonic antigen, MDT: multidisciplinary team, RECIST: Response Evaluation Criteria in Solid Tumors).

**Table 2 cancers-14-03952-t002:** Uni- and multivariate Cox regression analysis on factors influencing overall survival. As a standard criterion, values of hazard ratio (HR) below 1 are considered protective, while values above 1 are detrimental to survival.

VARIABLE		HR Univariable	HR Multivariable
**Age**	Mean (SD)	1.00 (0.99–1.01, *p* = 0.806)	-
**Site of primary tumor**	cecum	1.0	-
	ileum	0.59 (0.08–4.35, *p* = 0.602)	-
	ascending colon	1.13 (0.72–1.78, *p* = 0.582)	-
	transversal colon	1.04 (0.58–1.84, *p* = 0.903)	
	descending colon	0.76 (0.49–1.17, *p* = 0.208)	-
	rectum	1.05 (0.68–1.63, *p* = 0.823)	-
**T stage**	1	1.0	-
	2	1.12 (0.12–10.75, *p* = 0.923)	-
	3	2.72 (0.37–19.75, *p* = 0.322)	-
	4	2.41 (0.34–17.21, *p* = 0.380)	-
**N status**	0	1.0	-
	1	1.21 (0.93–1.56, *p* = 0.152)	1.11 (0.79–1.55, *p* = 0.554)
	2	1.66 (1.27–2.18, *p* < 0.001)	1.71 (1.20–2.43, *p* = 0.003)
**KRAS**	Wild type	1.0	-
	Mutated	0.77 (0.62–0.95, *p* = 0.016)	0.72 (0.54–0.96, *p* = 0.024)
**Systemic therapy versus loco-regional therapy**	Systemic therapy	1.0	-
	Locoregional treatment	0.32 (0.26–0.39, *p* < 0.001)	0.21 (0.14–0.29, *p* < 0.001)
**Bilobar disease**	No	1.0	-
	Yes	1.79 (1.47–2.18, *p* < 0.001)	1.22 (0.86–1.72, *p* = 0.265)
**Synchronous vs. Metachronous**	synchronous	1.0	-
	metachronous	0.84 (0.69–1.02, *p* = 0.084)	-
**Size of metastases**	Mean (SD)	1.04 (1.03–1.06, *p* < 0.001)	1.11 (1.06–1.17, *p* < 0.001)
**Number of metastases**	Mean (SD)	1.02 (1.01–1.03, *p* < 0.001)	1.00 (0.98–1.02, *p* = 0.842)
**CEA (ng/mL)**	Mean (SD)	1.00 (1.00–1.00, *p* < 0.001)	1.00 (1.00–1.00, *p* = 0.050)
**Time to treatment**	Mean (SD)	1.00 (1.00–1.00, *p* = 0.076)	-
**Order of resection**	Simultaneous resection	1.0	-
	Liver first	0.94 (0.58–1.53, *p* = 0.817)	-
	Bowel first	1.12 (0.78–1.61, *p* = 0.542)	-

**Table 3 cancers-14-03952-t003:** Baseline characteristics of patients who had systemic therapy (ST) or locoregional treatments (LRT).

	PRE-Propensity SCORE			POST-Propensity SCORE		
	ST	LRT	*p*	ST	LRT	*p*
**n**	250	542		209	209	
**SEX = male (%)**	171 (68.4)	340 (62.7)	0.15	144 (68.9)	131 (62.7)	0.216
**Age (median [IQR])**	63.00 [54.00, 71.75]	63.00 [55.00, 70.00]	0.754	63.00 [55.00, 72.00]	61.00 [53.00, 70.00]	0.048
**T stage (%)**			<0.001			0.245
**0**	1 (0.4)	4 (0.7)		7 (3.3)	2 (1.0)	
**1**	1 (0.4)	14 (2.6)		12 (5.7)	7 (3.3)	
**2**	15 (6.0)	68 (12.6)		24 (11.5)	24 (11.5)	
**3**	115 (46.0)	324 (59.8)		108 (51.7)	105 (50.2)	
**4**	36 (14.4)	94 (17.3)		58 (27.8)	71 (34.0)	
**NA**	82 (32.8)	38 (7)				
**N stage (%)**			<0.001			0.747
**0**	48 (19.2)	152 (28.1)		95 (45.5)	91 (43.5)	
**1**	60 (24.0)	208 (38.4)		59 (28.2)	56 (26.8)	
**2**	60 (24.0)	140 (25.8)		55 (26.3)	62 (29.7)	
**NA**	82 (32.8)	42 (7.7)				
**KRAS WT (%)**	89 (46.6%)	246 (54.3%)	0.089	128 (61.2%)	119 (56.9%)	0.426
**Metachronous disease (%)**	78 (31.2)	214 (39.5)	0.029	72 (34.4)	70 (33.5)	0.918
**Bilobar disease (%)**	173 (69.2)	297 (54.8)	<0.001	133 (63.6)	137 (65.6)	0.759
**Extrahepatic** **disease (%)**	93 (37.2)	138 (25.5)	0.001	69 (33.0)	74 (35.4)	0.68
**Size of the largest metastasis (cm)**						
**Median [IQR]**	3.50 [2.20, 5.60]	3.00 [1.80, 4.50]	<0.001	3.20 [2.00, 5.20]	3.30 [1.80, 5.40]	0.967
**N of metastases**						
**Median [IQR]**	5.00 [2.00, 10.00]	4.00 [2.00, 7.00]	<0.001	5.00 [2.00, 10.00]	5.00 [2.00, 10.00]	0.671
**CEA (ng/ml)**						
**Median [IQR]**	19.00 [5.00, 69.50]	5.60 [3.00, 18.00]	<0.001	8.60 [4.00, 16.00]	7.20 [3.00, 18.00]	0.718
**Time to treatment (months)**						
**Median [IQR]**	17.00 [7.00, 27.00]	30.00 [15.00, 55.00]	<0.001	8.00 [6.00, 13.00]	20.00 [8.00, 48.00]	<0.001

## Data Availability

The retrospective analysis was performed by using data of the adult patients enrolled in the liver unit registry and was conducted according to the guidelines of the Declaration of Helsinki. The data sets generated and/or analyzed during the current study are not publicly available but are available from the corresponding author on reasonable request.

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
