# Peer review of "Multidisciplinary Tumor Board in the Management of Patients with Colorectal Liver Metastases: A Single-Center Review of 847 Patients"

_cancers, 2022, doi:10.3390/cancers14163952_

Round 1

Reviewer 1 Report

Thank the authors for their efforts to revise the manuscript. This paper is now more acceptable for me.

There is one point to be clarified.

In table 3 (comparison of the backgrounds between ST and LRT group), there is large differences in CEA values between the groups before and even after PSM. If there is no differences in extrahepatic mets and other tumor conditions  (My understanding is "etratahepatic disease" and other factors without differences in the table after PSM means that), do the authors allocate those patients to ST only due to high CEA value during MDT evaluation plus something additional reasons? If the authors can make clarifying detailed description of this in discussion section, it might be supportive explanation of the advantages of MDT approach for specific patients' group.  Also, how many percentage of patients in LRT group did get peri-operative (combination) chemotherapy should be clarified. 

Author Response

We thank Reviewer 1 for these observations. Please note that we have now fixed that scribal error about CEA. Since CEA level was significantly different between the two groups (ST group vs. LRT group), it together with all the other statistically significant different factors, it was used to build the matched cohort. We are sorry about that, but again in the matched cohort there was no difference among tumoral factors. We have also modified Figure 1 to make the reader better understand the subgroups of patients included in the study. This change in Figure 1 should answer your last question. Thank you.

Reviewer 2 Report

The authors have obviously made a great effort this time to modify the paper according to my comments and those of the academic editor. I thank them for their efforts and believe the manuscript is now significantly improved compared to the original version. The tone is softer regarding any conclusions drawn, certain comparisons have been dropped, and the paper is more descriptive. I believe the manuscript can be accepted for publication in its current form, pending spelling / grammar corrections from a native English speaker.

Author Response

We thank Reviewer 2 for this note. We have just now revised the paper for English language. However, please be advised that Cancers should make its own revision by using a native English speaker. Thank you.